# Analysis and Optimization of Mn Removal from Contaminated Solid Matrixes by Electrokinetic Remediation

**DOI:** 10.3390/ijerph17061820

**Published:** 2020-03-11

**Authors:** Claudio Cameselle, Susana Gouveia, Adrian Cabo

**Affiliations:** BiotecnIA, Chemical Engineering, University of Vigo, Rua Maxwell s/n, Building Fundicion, 36310 Vigo, Spain; gouveia@uvigo.es (S.G.); acabo@uvigo.es (A.C.)

**Keywords:** electrokinetic remediation, manganese, pH control, citric acid, kaolinite, sludge

## Abstract

Electrokinetic remediation is a useful technique for the removal of ionic contaminants in soils, sediments, sludges, and other solid porous matrixes. The efficiency of metal removal and the electricity consumption in the electrokinetic treatment of soils largely depend on electric and physicochemical conditions. This study analyzes the electrokinetic treatment of Mn contaminated kaolinite clay specimen and the influence of voltage, current intensity, moisture content, pH, and facilitating agents on metal removal and energy consumption. The objective of this study is to identify the influence of the typical variables used in electrokinetic remediation. The results showed that the operation at constant voltage or constant current intensity were equivalent in terms of metal removal and energy consumption, as long as the electric field intensity was kept low to minimize the consumption in parallel electrochemical reactions, especially the electrolysis of water. The moisture content had a significant influence on the Mn removal. Moisture content higher that 50 percent resulted in very effective Mn removal as compared with kaolinite specimens with lower moisture. The control of pH in the electrolyte solutions and the addition of facilitating agents (organic acids) enhanced the removal of Mn but increased the electric energy cost. Overall, the best conditions for Mn removal involved low to moderate electric potential difference (10 to 30 V), the use of citric acid as the facilitating agent, and the pH control in the cathode at a slightly acid pH. The electrokinetic treatment of a sludge from a water treatment plant contaminated with Mn was effective when pH control on the cathode was used. Mn and various metals (66% of Mn, 30% of Cu, 56% of Zn, 21% Sr, and 21% of Fe) were removed with moderate electricity and acid consumption.

## 1. Introduction

Electrokinetic remediation is a soil decontamination technology extensively studied from the 1990s in the twentieth century. The electrokinetic treatment of contaminated soils is based on the application of a low intensity electric field directly to the soil with two main electrodes, i.e., anode and cathode. The electric field induces the mobilization of the contaminants and their transportation towards the electrodes. The contaminants are removed by two main mechanisms, electromigration (the transport of ions towards the electrode of opposite charge) and electroosmosis (a net flux of water and interstitial fluid induced by the electric field, the interstitial fluid, flows towards the cathode in electronegatively charged soils). Furthermore, the electrochemical reactions induced by the electric field on the electrodes has an enormous influence in the speciation of the contaminants and their transportation out of the soil. The main reactions induced by the electric field are the electrolysis of water that generates H+ ions in the anode and OH− ions in the cathode. This reaction and the pH changes induced by the H+ and OH− ions are decisive in the efficiency of electrokinetics for the removal of metals from contaminated soils [1].

In 2009, Reddy and Cameselle [2] edited a comprehensive book with the fundamentals and applications of electrokinetics, with contributions from renowned researchers around the globe. Despite all that research and published studies for about 30 years, electrokinetics is still being studied for the removal of various organic and inorganic contaminants in soils, sediments, sludge, and other solid porous matrices [3,4]. There are many studies in the literature at the lab scale, but the applications at the field scale are still limited to the work of Lageman [5,6], Electrokinetics Inc. [7], Electropetroleum Inc. [8], Electrokinetic Ltd. UK [9], Monsanto [10], and the projects by USEPA [11] and USDOE [12]. Probably, the short number of commercial applications at field scale is due to the limited understanding of the electrokinetic phenomena and the relation among the physicochemical properties of the soil, the contaminants, and the electric field [13,14].

The electrokinetic remediation has been studied for the removal of heavy metals and toxic elements (Cu, Cd, Zn, As, etc.) [15,16], other inorganic contaminants (F^−^, NO_3_^−^, etc.) [17,18], and various organic contaminants (hydrocarbons, pesticides, Polycyclic Aromatic Hydrocarbons (PAHs), etc.). The removal of organic contaminants requires their solubilization using various facilitating agents [19,20] such as the following: co-solvents miscible with water, surfactants, cyclodextrins, or any other compound that solubilizes the target contaminant with minimum negative effect for the environment. As an example, various surfactants have been proposed for the solubilization of PAHs from contaminated soil [21], the electrokinetic treatment was only successful when the surfactant solubilized the organic contaminant, whereas the operating conditions maintained a high electroosmotic flow. Heavy metals have been the target contaminant of many electrokinetic remediation studies [22]. The effective removal of metals from soil by electrokinetics depends on their speciation and solubilization. Thus, the pH of the soil is one of the main parameters to consider in any electrokinetic application [23]. As it was explained before, the pH of soil is affected by the electrolysis of water induced by the electric current. Thus, the metals are solubilized in the anode side and accumulated in the cathode side because of the high pH conditions. Researchers mainly used pH control on the cathode [24] and the addition of complexing agents [25] to keep the metals in solution to assure their transportation in the electric field, but they payed less attention to electric and electrochemical aspects that also have an important influence in the effective removal of metals by electrokinetics [26].

This work focuses on the assessment of the influence of the electric field strength and its mode of application to the soil in the electrokinetic removal of heavy metals. In order to understand the influence of the electric parameters, a simple model soil composed of kaolinite clay contaminated with manganese sulfate was selected for this work. This model soil simplified the geochemistry of the soil for the identification of the influence of the electric field on physicochemical parameters (pH, electric conductivity, ion concentration, and moisture content), and on the transportation of the contaminant metals. In addition, the electric energy consumption and the treatment time were considered in the study as fundamental parameters for the design of practical applications and large scale. The Mn results and conditions obtained in the tests with model soil samples are used for real samples of an industrial sludge contaminated with Mn. The electrokinetic tests with model samples could help in defining the operating conditions of the real samples. However, the electrokinetic remediation of real samples requires further study and optimization since the geochemistry and aging of contaminants is different than the model samples and they will affect the results [27]. This study was based on the previous work by Ricart et al. [28,29] who studied the electrokinetic removal of manganese in model and real samples, because of the impact of such metal in the surroundings of a former lignite mine in Galicia (NW Spain). Thus, the objective of this study is to advance the identification of conditions for the development of a practical application for the removal of Mn in soils.

## 2. Materials and Methods

### 2.1. Electrokinetic Cell

The electrokinetic cell used in all the tests was designed in our research group based on a previous design by Pamucku et al. [9] from the University of Lehigh, PA, USA. The electrokinetic cell (Figure 1) was composed of a central tube and two electrode compartments. The central tube was 10 cm long and 32 mm in diameter and it held the kaolinite specimen. At both ends of the central tube, the two electrode compartments held the main electrodes (anode and cathode) immersed in an electrolyte solution (anolyte and catholyte). Two porous stones and two glass fiber filter disks between the central tube and the electrode compartments retained the soil specimen, avoiding any leak into the electrolyte solutions. The volume of the electrode compartments was 300 mL, and they were filled with DI water or the solution indicated in each experiment. Two disks of graphite of 50 mm diameter were used as main electrodes (anode and cathode) connected to a DC power supply. Three auxiliary electrodes (namely P1, P2, and P3 from the anode to the cathode) in the central tube were used to measure the electric gradient along the soil specimen. The flow control panel was used to measure the electroosmotic flow through the soil specimen [30].

### 2.2. Consolidation Unit

Figure 2 shows a diagram of the consolidation unit designed by our research group and constructed at the University of Vigo. This unit was used to create moisture saturated and homogeneous soil samples from a slurry poured into the electrokinetic tube. The consolidation unit was able to apply small increments in the pneumatic stress to the soil or kaolinite slurry while allowing the excess of fluid from the sample to drain out [28].

### 2.3. Model Soil Preparation

The model soil specimen used in all the tests was composed of kaolin contaminated with manganese sulfate. Kaolin, supplied by the company CAVISA (Vimianzo, A Coruña, Spain), has an average particle size of 3 μm and a specific surface area of 13.5 m^2^/g. Mineralogy analysis by X-ray diffraction showed the presence of 85% kaolinite clay, 14% mica, and 1% quartz. The particle size was selected based on the availability of kaolin commercial products from the company CAVISA. The smallest size (3 μm) was selected to simulate the behavior of a low permeability clayey soil. Kaolinite was selected since it shows much lower buffering capacity and cation-exchange capacity as compared with other clay minerals. The model soil specimen was prepared mixing 100 mL of concentrated manganese sulfate solution with 120 g of dry kaolinite. The mixture stood overnight (about 24 h) before it was used in the electrokinetic tests. The final mixture had a moisture content of 45% and a concentration of 5 g Mn/kg of dry kaolinite.

The spiked kaolinite was introduced in the central tube of the electrokinetic cell using the consolidation unit. This consolidation unit created homogeneous and saturated soil specimens for the electrokinetic tests. During the consolidation process, a fraction of water and manganese was lost. The final Mn concentration in the kaolinite specimen ranged from 3.6 to 3.8 g Mn/kg of dry soil and the moisture content ranged from 37% to 40%. The initial pH of the contaminated kaolinite was pH = 4 to 4.5.

### 2.4. Contaminated Sludge from an Industrial Area

A sludge from a water treatment plant in an industrial area was used to test the capability of electrokinetic remediation to remove metals from real solid matrixes. The sludge was contaminated mainly with Mn, although it contained significant amounts of other metals (Table 1).

### 2.5. Testing Procedures

The contaminated kaolinite with Mn or the industrial sludge was introduced in the central tube of the electrokinetic cell and it was compacted in the consolidation unit. Then, the electrode chambers were attached to the central tube and filled with deionized (DI) water. The main electrodes were connected to the direct current (DC) power supply and the electrode chambers were connected to the flow panel. A constant DC electric potential (10 to 30 V) or electric current intensity (1.2 mA) was applied through the main electrodes for a maximum of 7 days (except some specific tests which duration is specified in the results section). Periodic readings of electric current intensity and electric potential among the main and auxiliary electrodes were recorded. The electroosmotic flow was measured by volume variations in the flow panel. The pH of the electrode solutions was measured along the testing time and registered. At the end of the test, the electrode solutions were collected and analyzed for pH and metal concentration. The kaolinite specimen (or the industrial sludge) was extruded from the central tube and divided into 5 equal portions or sections, namely S1 to S5 from anode to cathode. A sample of the initial kaolinite specimen (or the industrial sludge) and the samples from the 5 final portions were analyzed for metal concentration, pH, and moisture content.

### 2.6. Analytical Methods

Moisture content in kaolinite or sludge samples was determined by dry weight at 105 °C until constant weight. The pH in the solid samples was determined mixing 1 g of solid and 2.5 mL of DI water for 1 h, and then the pH was measured in the supernatant fluid. The Mn in the liquid and solid samples was determined by inductively coupled plasma optical emission spectrometry (ICP-OES) with a piece of equipment Optima 4300 DV from Perkin Elmer (Waltham, Massachusetts, USA). The samples were firstly digested following the USEPA method 3010A (acid digestion of aqueous samples and extracts for total metals for analysis by FLAA or ICP spectroscopy) or method 3050B (acid digestion of sediments, sludges, and soils).

## 3. Results

### 3.1. Electrokinetic Treatment at Constant Electric Potential

The electromigration of Mn in contaminated kaolinite specimen was firstly tested at a constant electric potential gradient of 30 V. This value was selected based on previous electrokinetic tests with metal contaminated kaolinite [8,28] where it was found that 3 DcV/cm was an appropriate voltage gradient for effective removal of metals. As it can be seen in Figure 3a, Mn was transported towards the cathode by electromigration and accumulated in the last section of soil (S5), close to the cathode. The average removal of Mn in sections S1–S4 was 83%. Mn accumulated in section S5 (69%) and in the cathode chamber (14%) due to the high pH in the cathode side. The amount of Mn in the anode was negligible. The moisture content was slightly lower than the initial moisture due to the electroosmotic flow that generated a uniform flow of water from anode to cathode (Figure 4a) [21].

The pH and Mn distribution in kaolinite specimen at the end of the experiment can be explained based on the pH changes induced by the electrolysis of water in the electrode chambers. The oxidation of water in the anode yielded H^+^ ions, and the reduction of water in the cathode yielded OH^−^ ions. Thus, the anode was acidified and the cathode was alkalinized (Figure 4b). The acidification in the anode and the penetration of the acid front in the kaolinite specimen decreased the pH of the solid matrix (Figure 3c), desorbing metals, i.e., Mn^2+^, and other ionic species that migrated towards the cathode due to the effect of the electric field. The manganese was transported as Mn^2+^ towards the anode until reaching a zone of high pH (Section S5, Figure 3a), where the Mn^2+^ ions were immobilized as manganese hydroxide. This zone of high pH in Section S5 was due to the electromigration of OH^−^ ions from the cathode. The manganese (II) hydroxide, Mn(OH)_2_, showed a white-yellowish color, but this compound easily transformed into a mixture of oxides and hydroxides with the general formula MnO(OH) with a characteristic brown color [30]. Thus, the precipitation of Mn was clearly observed in the electrokinetic cell as a brown band in the white kaolinite or as a precipitate in the cathode chamber.

The pH changes in the electrode chambers, the mobilization of Mn^2+^, and the precipitation of manganese in Section S5 explain the electric current intensity profile and the distribution of the potential gradient in the system (Figure 5a,b). At the beginning of the test, the electric current intensity was low because there were no ions in the liquid in the electrode chambers. The electrolysis of water and the mobilization of the Mn^2+^ ions increased the number of ions available to transport the electric current. The current intensity increased up to a maximum of 2.3 mA and, then, decreased due to the immobilization of Mn^2+^ ions for the alkaline environment in the cathode side. The high pH in the cathode was responsible for the concentration of the voltage gradient in the cathode (Figure 5b) and the increase of electric resistance in the cathode solution (resistance in P3-Cathode, Figure 5d) and the section of soil close to the cathode (resistance between the auxiliary electrodes P2 to P3, Figure 5c). The increase of electric resistance matched the decrease of the current intensity (Figure 5a,c,d).

The electric power consumption of the electrokinetic treatment in seven days was 7.18 Wh. About one third of the electricity was spent in the electrolysis of water. The extension of the electrolysis of water was determined by registering the gas (H_2_) produced at the cathode. The amount of electricity consumed in the electrolysis of water is compared with the total charge that passed through the electrokinetic cell in terms of mol of electrons (Figure 6).

Overall, the electrokinetic treatment of a Mn contaminated kaolinite specimen was able to remove 83% of initial manganese from Sections S1–S4 and the Mn was concentrated in the last section of kaolinite (S5) and the cathode chamber.

### 3.2. Electrokinetic Treatment at a Constant Electric Current Intensity

The operation at a constant electric potential or electric current intensity are the two alternative modes for any electrochemical test. The operation at a constant electric potential difference between anode and cathode implies that the driving force for the transportation of ions is constant. Thus, the total flux of ions, i.e., the total charge through the system, varies along the treatment time depending on the availability of ions, in other words, the electric conductivity of the system. The operation at constant electric current intensity means that the flow of charge, i.e., the flow of ions, is constant, and the voltage gradient varies depending on the availability of ions to transport the charge between anode and cathode.

The test at a constant electric current intensity was carried out at 1.5 A/m^2^ (1.2 mA in the experimental cell used in this study). This is an intermediate value of the current intensity registered in the previous test at 30 V. This value was selected to be able to compare the results of this test at a constant current intensity and the previous test at a constant 30 V. The results of Mn removal and electric power consumption are shown in Figure 7. The operation at a constant current intensity resulted in higher removal of Mn in the cathode, and less accumulation in Section S5. However, the removal of Mn from Sections S1–S4 was only 65%. The use of a relatively low intensity resulted in the development of a relatively low electric potential between the anode and cathode. As a result, the extension of the electrolysis of water was lower and the development of the alkaline zone in the cathode side required more time than in the test at a constant 30 V. In these conditions, more Mn was able to electromigrate towards the cathode and accumulated in the cathode chamber before the development of the alkaline environment. When the pH was high enough to precipitate the metal, the Mn^2+^ ions started to accumulate in Section S5. The lower removal of Mn in Sections S1 to S4 was due to the lower driving force (electric potential difference) developed in the system because of the use of such a low electric current intensity. This was also the reason for the lower electric power consumption.

Overall, the response of the electrokinetic system seemed to be similar at a constant current intensity or a constant electric potential. It is important to keep the electric field intensity low to decrease the electricity spent in the electrolysis of water. Low voltage or current intensity would result in low energy consumption, but the removal of metal would require more time.

### 3.3. Treatment Time

The profile of the electric current intensity in Figure 5a (electrokinetic test at 30 V) showed a peak at around three days and, then, the intensity decreased to a very low value, below 1 mA in seven days. A low current intensity means that there was no transport of ions, and therefore it is assumed that there was no removal of Mn when the current intensity was very low. To elucidate the influence of the treatment time and the electric current intensity on the removal of Mn, four tests were designed and run at 3.5, 6, 7, and 14 d. The profile of the current intensity of the four tests is plotted in Figure 8. Table 2 shows the Mn removal, Mn mass balance at the end of the tests, and the electric power consumption.

As can be seen in Table 2, the operation of the electrokinetic test until the peak of current intensity (Test T1) showed a low removal of Mn, whereas Tests T2 and T3 (operated for six or seven days) clearly increased the removal of Mn from soil with a moderate increment in the electric power consumption. These tests were stopped when the electric current intensity reached a value about 1 mA. Test T4 was operated for 14 days but the Mn removal was very similar to Test T3. However, the operation for 14 days increased the power consumption with no increment in metal removal. Overall, the operation of this electrokinetic treatment can be stopped based on the actual value of the electric current intensity. The results suggest a limit value of 1 mA (1.2 A/m^2^).

### 3.4. Effect of the Electric Potential Difference

The electric potential difference used in the electrokinetic tests could have a significant influence on the removal of Mn and electric power consumption. Voltage increases the driving force for the transportation of cations (Mn^2+^) from the anode to cathode, but at the same time, an increase in voltage also increases the total electric power consumption due to the major extension of parallel reactions, i.e., the electrolysis of water. As it can be seen in Table 3, the higher removal of Mn from soil was observed at 30 V, and the fraction of removed Mn decreased with the voltage. At the same time, the total power consumption sharply decreased with lower voltage. These results suggest that there is a tradeoff between treatment time and voltage. The electrokinetic treatment requires shorter time at high voltage, but the energy cost is much higher. Test V5 (Table 3) is a good example of the influence of voltage. The removal of Mn in Test V5 was over 90% with very low power consumption, but the treatment required 40 days. Additionally, it is important to note that the decrease of the voltage also favored the accumulation of Mn in the cathode, decreasing the Mn in the S5 soil section. This effect is due to the lower extension of the electrolysis of water at lower voltage. Thus, the alkaline environment in the cathode took more time to develop and Mn ions had more time to electromigrate into the cathode solution. Overall, it seems that low voltage is more effective for the removal of Mn with low energy consumption, but the treatment requires much more time.

### 3.5. Effect of the Kaolinite Specimen Moisture Content

The removal of metals from soil by electrokinetic remediation is based on the mobilization and solubilization of the metals in the interstitial fluid. Then, the metals can be transported out of the soil by electromigration or electroosmosis. Thus, the moisture content of the soil specimen is critical to achieve an effective removal of the metal. The moisture of the kaolinite specimen in the electrokinetic cell (Figure 1) was adjusted from 40% to 60% and five tests were run to evaluate the effect of the moisture on the removal of Mn (Table 4). No higher or lower moisture contents were tested because the kaolinite sample was too dry or too fluid to operate in the electrokinetic cell. Test W1 with 40% moisture resulted in 83% of the removal in Sections S1–S4. Mn was accumulated in Section S5 (69%) and cathode chamber (14%). The increase of the moisture content (Tests W2–W5) clearly favored the electromigration of Mn towards the cathode and no accumulation of Mn was observed in Section S5. The residual Mn in the kaolinite specimen decreased with moisture content, and the accumulation of the metal in the cathode chamber increased, up to 95% at 60% moisture. The results showed that 60% moisture was an appropriate value to remove the Mn in the model kaolinite samples. The electric power consumption also increased with the moisture content because the presence of water decreased the resistance of the soil sample to the electrokinetic transport of charge.

### 3.6. pH Control in the Cathode

The main limitation in the removal of Mn from the kaolinite specimen was the development of an alkaline environment in the cathode side that precipitated the Mn in the last section of kaolinite, Section S5 (Figure 3a). The simplest solution to avoid the premature precipitation of Mn is the neutralization of the alkaline environment in the cathode with the controlled addition of an acid [31,32]. In Tests H2 to H6, sulfuric acid was used to depolarize the reduction of water in the cathode and keep an acid environment at the selected pH values, as shown in Table 5. Sulfuric acid was selected because the ion sulfate does not interact with the target cation, Mn^2+^. In the depolarization of the reduction of water it is important to select an acid that does not form insoluble salts with the contaminated metals, because the anion of the acid is transported into the soil by electromigration and forms metal precipitates in the soil, limiting the efficiency of the treatment.

As shown in Table 5 and Figure 9, the neutralization of the alkaline environment in the cathode (Tests H2–H6) avoided the premature precipitation observed in Test H1 with no pH control. Mn was completely removed from the kaolinite specimen in the tests at seven days (Tests H2 and H4), but the tests at four days (H3 and H5) also showed a very high removal, over 85%. In general, the lower the pH, the higher the removal, but the differences in removal among the tests are not as important as the electric power consumption. Lower pH values increased the conductivity of the system and the energy consumption sharply increased. Such high values of power consumption were not due to the transportation of Mn but to the larger extension of parallel reactions, i.e., the electrolysis of water. Thus, the most appropriate pH to achieve a complete removal with moderate electric power consumption is an acid pH close to neutrality (pH = 6 to 7).

### 3.7. Enhanced Remediation with Facilitating Agents

An alternative method to overcome the premature precipitation of Mn in the last section of kaolinite S5 (Figure 3a) is the use of complexing agents. Various organic acids form stable complexes with heavy metals, including Mn [33]. These complexes are stable in a wide range of pH and can be soluble at alkaline pH, allowing their electrokinetic transport into the cathode chamber. Various authors have tested and discussed the use of complexing agents in the enhanced removal of metals in electrokinetic remediation [25,34,35]. Wong et al. [35] described the chemical equilibrium of the EDTA in solution in the presence of selected metallic cations. The predominant species, ionic metal, or complexed metal, depended on the chemical affinity metal ligand (EDTA) and the medium pH. EDTA delivered in the cathode chamber electromigrated into the soil specimen dissolving Pb and Zn from soil, forming negatively charged complexes that were completely removed into the anodic solution. Cameselle and Pena [36] studied the removal of Zn form an agricultural contaminated soil using citric acid as the facilitating agent. The predominant species in the system, Zn citrate (L, ligand citrate), was the negative complex ZnL^−^ at pH > 6; the neutral complex ZnHL at pH 3.5, and the positive complex ZnH_2_L^+^ at pH 3. At pH below 3, the predominant species was the cation Zn^2+^. The citric acid delivered to the soil from the electrode chambers was able to dissolve the contaminated Zn, and then it was transported towards the anode or the cathode (by electromigration or electroosmosis) depending on the soil pH and the charge of the predominant species. The literature results proved the efficiency of the complexing agents removing metals from real soil samples, but the results were very affected by the pH.

In this study, the removal of Mn from model kaolinite specimens was tested using five organic acids with complexing capacity, i.e., acetic acid, EDTA, citric acid, tartaric acid, and oxalic acid (Table 6). The organic acids were added at the concentration 0.1 M to the manganese solution before mixing it with kaolinite. The final pH of the kaolinite mixture ranged from pH 3 (for oxalic acid) to pH 5 (for EDTA). The pH of the mixture defined the possible complexes formed between the Mn^2+^ ions and the organic acid. In the pH range used in these tests, the formation of anionic complexes is not favored, the most probable complex species are neutral or cationic [33,36]. Thus, it was expected that the Mn would be transported towards the cathode by electromigration (for cationic species) and electroosmosis (for neutral species). The results in Table 6 and Figure 10 shows that the use of organic acids favored the transportation of Mn towards the cathode decreasing the accumulation in Section S5 of the kaolinite specimen. Thus, in Tests F2 to F6 the amount of Mn accumulated in Section S5 was lower than in Test F1 (with no facilitating agent) and the Mn was accumulated in the cathode chamber. As expected, the better results with facilitating agents were obtained at higher power consumption. The higher energy expenditure was mainly due to the high conductivity of the system because of the higher ionic concentration in the kaolinite. It is important to note the results with citric acid. Mn was completely removed from the soil and accumulated in the cathode chamber. These results were due to the combination of the complexing capacity of citric acid with Mn and the development of an intense electroosmotic flow that was able to transport all the soluble species in the interstitial fluid towards the cathode. As reported in the literature [37,38], citric acid interacts with the soil particles enhancing the development of a high electroosmotic flow. Overall, the treatment with citric acid as the facilitating agent is believed to be the most appropriate for metal removal among the various facilitating agents tested.

### 3.8. Treatment of Mn Contaminated Sludge

The electrokinetic treatment was applied to a sludge from a water treatment plant in an industrial area. The plant treated the runoff water from a mining area by neutralization and precipitation of metals in solution and particles in suspension. The sludge contained a significant amount of metals and carbonates with a pH of 7.60. In this condition, it seemed that the best electrokinetic treatment should involve the depolarization of the reduction of water in the cathode (to suppress the alkaline environment in the cathode) and favor the acid front generated in the anode (Test EK2 in Table 7). Thus, the acid front acidified the sample and mobilized the metals that then can electromigrate towards the cathode. Sulfuric acid was used to depolarize the reduction of water in the cathode. Another test (EK1) with no depolarization of the reduction of water was run as a control experiment.

As can be seen in Table 7 and Table 8, the electrokinetic treatment was able to remove a significant amount of Mn (66% in EK2) and other metals (Fe, Sr, Mg, Cu, Zn, and Ca). The control Test EK1 only removed minor amounts of metals confirming that acidification of the sludge specimen is necessary to mobilize the metals. An acid leaching test with 1 molar HCl defined the maximum extension of the solubilization of metals by acidification (Table 8). The non-dissolved fraction by acid leaching corresponds to the metals in the crystalline structure of the minerals, and therefore fraction cannot be considered as a contaminant for being immobilized in the mineral structure. The comparison of the EK2 with the acid leaching confirmed that there was still room for improvement in the electrokinetic treatment. The results suggest that an increase in the treatment time in Test EK2 results in better removal of metals until reaching the removal defined by the acid leaching in Table 8. The electric power consumption in Test EK2 is high as compared with Test EK1 and the electrokinetic tests with kaolinite. However, the cost of energy (2.94 kWh/kg of sludge) is not especially high as compared with other remediation alternatives. The consumption of acid (3.46 mol H^+^/kg of sludge) to depolarize the cathodic reduction of water must be considered in the design of an electrokinetic treatment at a large scale.

## 4. Conclusions

The electrokinetic removal of Mn from model kaolinite samples and real industrial sludge is effective under the following selected operating conditions:The application of a constant electric potential of 30 V to the kaolinite specimen for 7 days resulted in a removal of Mn as high as 83% in Sections S1–S4 (4/5 of the kaolinite specimen). Mn mainly accumulated in the last section of kaolinite (69%) and cathode (14%).The operation at constant voltage or constant current intensity yielded similar results in terms of Mn removal and electric power consumption.The operation at low voltage (10 V) for longer time (40 d) increase the removal of Mn and decreased the electric power consumption as compared with the test at 30 V for 7 days.The increase of the moisture content up to 60% resulted in over 90% Mn removal in the whole kaolinite specimen.The pH control in the cathode resulted in the complete removal of Mn from the kaolinite, but a significant increment in the electric consumption was observed. The increase in the electric consumption could be restrained using pH close to the neutrality, avoiding very acid pH in the cathode.The use of organic acids as facilitating agents improved the removal of Mn. The test with citric acid showed complete removal due to the formation of Mn complexes and the enhancement of the electroosmotic flow.The electric power consumption was very affected by the extension of parallel reactions, i.e., the electrolysis of water. Any modification of the electrokinetic system (moisture content, pH, facilitating agents, etc.) that increased the electric conductivity of the system resulted in an increase of electric power consumption.The electrokinetic treatment of an industrial sludge contaminated with Mn and other metals was only effective with the depolarization of the electrolysis of water in the cathode with sulfuric acid. This acid was selected because it does not interact with the target metal Mn.

Overall, the results of the electrokinetic test in model and real samples suggested the feasibility of the electrokinetic treatment for the removal of Mn from contaminated matrices.

## Figures and Tables

**Figure 1 ijerph-17-01820-f001:**
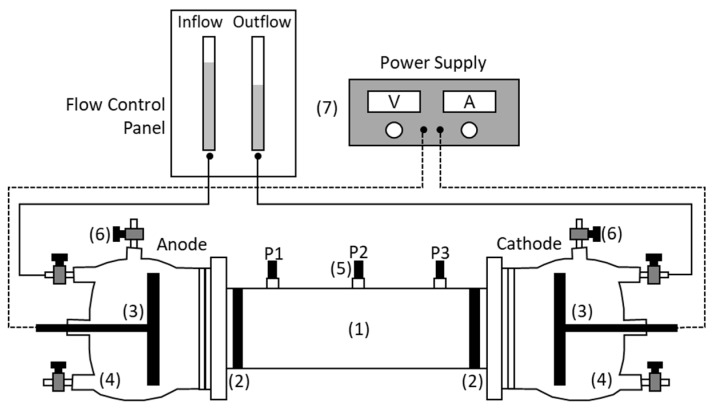
Electrokinetic cell and experimental setup. (**1**) Soil specimen; (**2**) Porous stones and glass fiber filter disks; (**3**) Main electrodes, anode and cathode; (**4**) Electrode compartments; (**5**) Auxiliary electrodes, namely P1, P2, and P3 from anode to cathode; (**6**) Gas vent valves; (**7**) DC power supply and panel for measuring the electroosmotic flow.

**Figure 2 ijerph-17-01820-f002:**
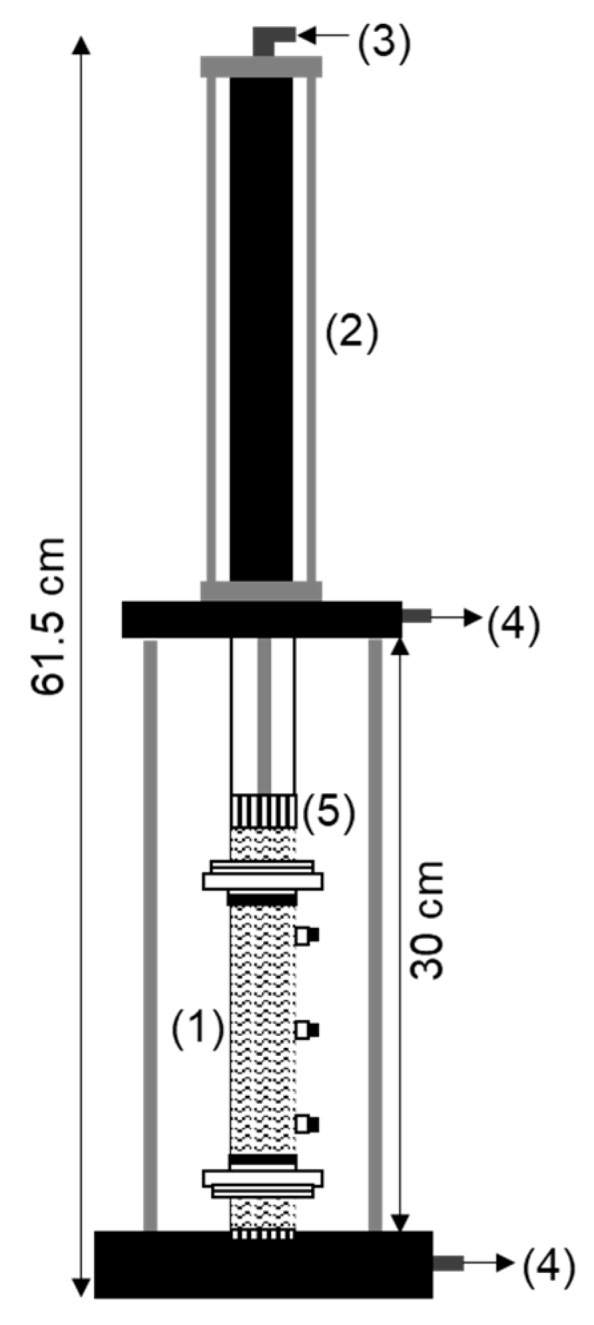
Diagram of the consolidation unit. (**1**) Central tube of the electrokinetic cell; (**2**) Pneumatic cylinder; (**3**) Pressurized air; (**4**) Water drainage; (**5**) Piston with a porous stone and a glass fiber filter disk.

**Figure 3 ijerph-17-01820-f003:**
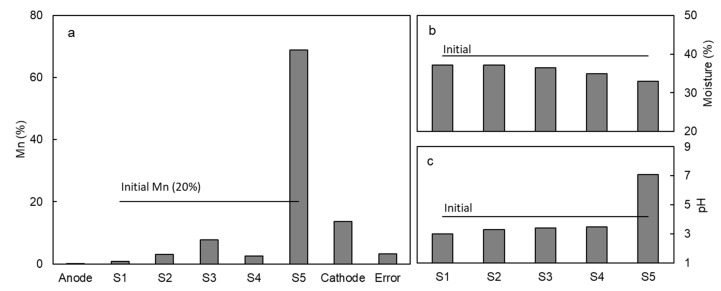
Mn concentration (**a**), moisture content (**b**), and pH (**c**) in the kaolinite specimen after the electrokinetic treatment at 30 V.

**Figure 4 ijerph-17-01820-f004:**
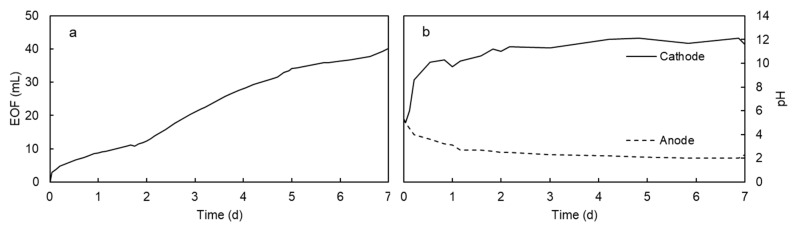
Electroosmotic flow (EOF) towards the cathode (**a**) and pH in the electrode solutions (**b**) in the electrokinetic test at 30 V.

**Figure 5 ijerph-17-01820-f005:**
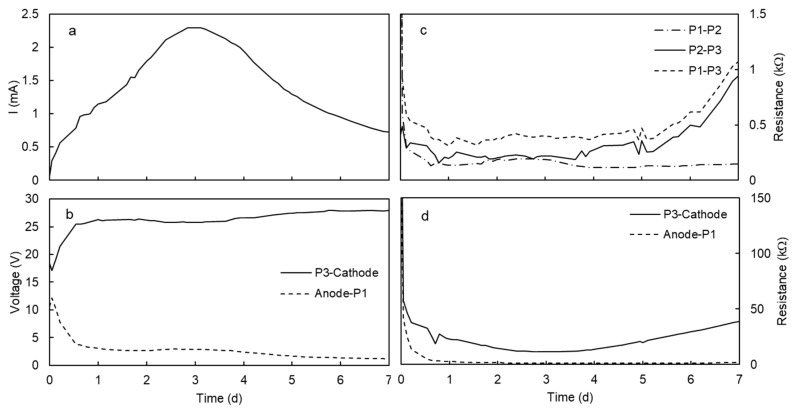
Electric current intensity (**a**), electric potential difference (**b**), and electric resistance in the kaolinite specimen (**c**) and the electrode chambers (**d**) during the electrokinetic test at 30 V. Voltage and electric resistance was determined using the main electrodes (anode and cathode) and the three auxiliary electrodes, namely P1, P2, and P3 from the anode to the cathode.

**Figure 6 ijerph-17-01820-f006:**
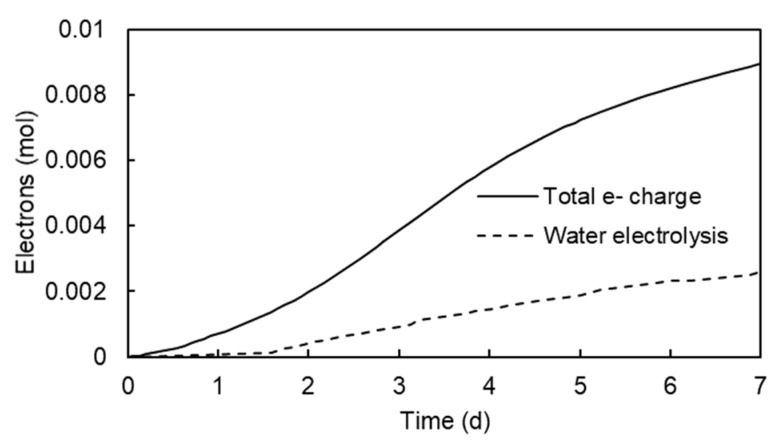
Electric power consumption in the electrokinetic test at 30 V and electricity consumed in the electrolysis of water.

**Figure 7 ijerph-17-01820-f007:**
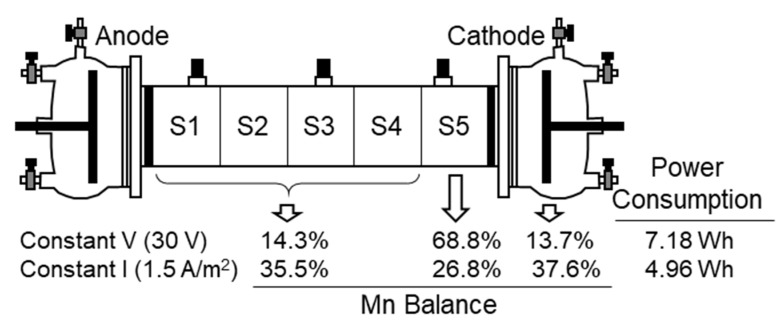
Comparison of the electrokinetic treatment at a constant electric potential or an electric current intensity.

**Figure 8 ijerph-17-01820-f008:**
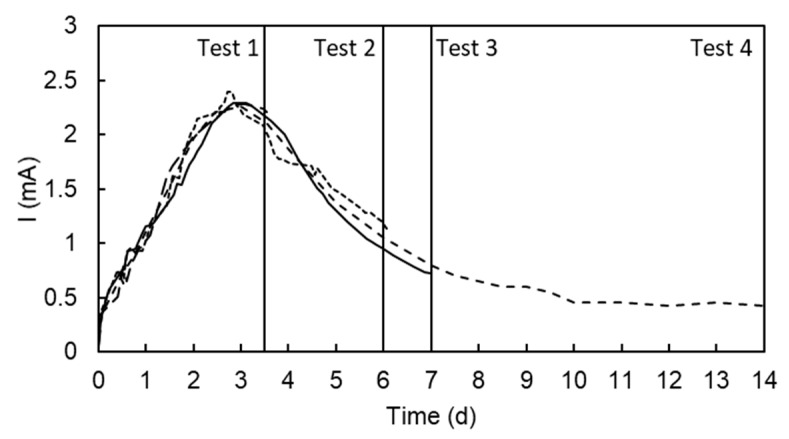
Electric current intensity of four electrokinetic tests at 30 V operated for 3.5 (Test 1), 6 (Test 2), 7 (Test 3), and 14 d (Test 4).

**Figure 9 ijerph-17-01820-f009:**
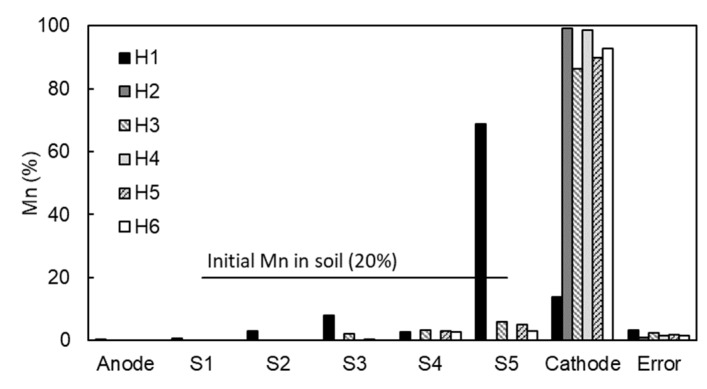
Mn distribution in the tests with pH control in the cathode (See conditions of tests in Table 5).

**Figure 10 ijerph-17-01820-f010:**
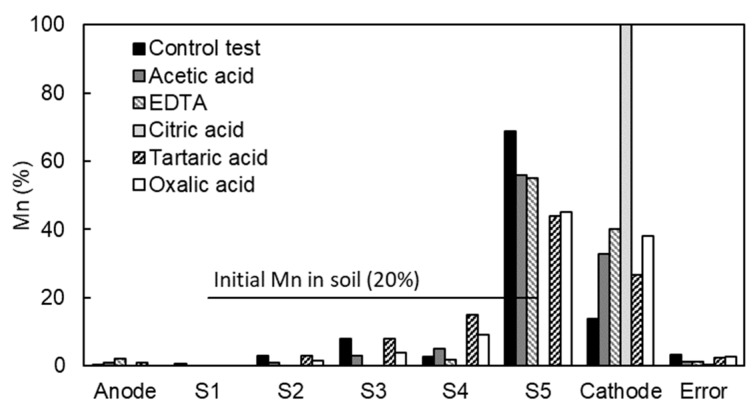
Mn distribution in the tests with organic acids as facilitating agents. Kaolinite specimen was divided into five sections namely S1 to S5 from anode to cathode.

**Table 1 ijerph-17-01820-t001:** Industrial sludge composition.

Element	Concentration (mg/kg)
Al	87323
Fe	39891
K	15440
Ca	10863
Ti	5661
Mg	4597
Na	6967
Mn	878
Zr	468
Sr	216
Zn	192
Cu	100

**Table 2 ijerph-17-01820-t002:** Treatment time effect on Mn removal and electric power consumption.

Test	Time (d)	Mn Removal in S1–S4 ^a^	Mn Mass Balance				Power Consumption (Wh)
			Mn in S1–S4 ^a^	Mn in S5 ^a^	Mn in Cathode	Error	
T1	3.5	64.06%	31.55%	51.81%	13.34%	3.19%	5.05
T2	6	76.9%	12.48%	70.08%	14.53%	2.86%	6.50
T3	7	82.92%	14.26%	68.84%	13.67%	3.20%	7.18
T4	14	84%	11.56%	70.5%	14.25%	3.64%	10.53

^a^ Kaolinite specimen sections, namely S1 to S5 from anode to cathode.

**Table 3 ijerph-17-01820-t003:** Influence of electric potential gradient in Mn removal and electric power consumption.

Test	Time (d)	Voltage (V)	Mn Removal in S1–S4 (%)	Mn Mass Balance (%)				Power Consumption (Wh)
				Mn in S1–S4	Mn in S5	Mn in Cathode	Error	
V1	7	30	82.92	14.26	68.84	13.67	3.20	7.18
V2	7	20	75.10	19.98	61.23	15.45	3.28	4.24
V3	7	15	70.50	28.36	50.74	18.34	2.53	2.68
V4	7	10	55.10	39.95	40.44	14.98	4.50	0.59
V5	40	10	90.98	7.28	66.64	20.91	4.90	2.76

**Table 4 ijerph-17-01820-t004:** Influence of the moisture content in Mn removal and electric power consumption

Test	Time (d)	Moisture Content (%)	Mn in Soil (%)	Mn in Cathode (%)	Error (%)	Power Consumption (Wh)
W1	7	40	83.10 ^a^	13.67	3.20	7.18
W2	7	45	33.66	64.00	2.33	11.98
W3	3	50	59.48	36.51	3.01	5.82
W4	7	50	24.38	72.02	3.58	15.76
W5	7	60	1.40	95.03	3.43	19.72

^a^ Residual Mn in Sections S1–S4 of kaolinite specimen.

**Table 5 ijerph-17-01820-t005:** Influence of pH control in the cathode on Mn removal and electric power consumption

Test	Time (d)	Cathode pH	Mn in Cathode (%)	Error (%)	Power Consumption (Wh)
H1	7	Alkaline	13.67	3.2	7.18
H2	7	6	99.12	0.88	55.2
H3	4	6	86.25	2.35	35.4
H4	7	4	98.56	1.44	85.6
H5	4	4	89.91	1.89	49.8
H6	4	2	92.76	1.52	95.3

**Table 6 ijerph-17-01820-t006:** Electrokinetic tests with organic acids (0.1 M) in Mn contaminated kaolinite specimen.

Test	Time (d)	Facilitating Agent	Mn in Soil (%)	Mn in Cathode (%)	Error (%)	Power Consumption (Wh)
F1	7	-	83.14	13.67	3.20	7.18
F2	7	Acetic acid	65.91	32.77	1.32	35.6
F3	7	EDTA	58.85	40.00	1.15	39.5
F4	7	Citric acid	0	99.99	0.01	32.3
F5	7	Tartaric acid	71.00	26.56	2.44	27.5
F6	7	Oxalic acid	59.42	38.00	2.58	43.2

**Table 7 ijerph-17-01820-t007:** Electrokinetic tests of a Mn contaminated industrial sludge.

Test	Voltage (V)	Time (d)	Mn in Sludge (%)	Mn in Cathode (%)	Power Consumption (Wh)	Consumed Acid (mol H^+^)
EK1	30	23.4	98.87	0.85	1.95	0
EK2	30	23.0	26.95	66.3	359	0.422

**Table 8 ijerph-17-01820-t008:** Metal removal from an industrial sludge by electrokinetics and acid leaching (HCl, 1 M).

Test	Metal Removal or Solubilization (%)											
	Al	Fe	K	Ca	Ti	Mg	Na	Mn	Zr	Sr	Zn	Cu
EK1	2	1	-	4	3	-	3	-	7	3	4	-
EK2	8	21	3	61	1	30	-	66	-	21	56	30
Acid leaching (HCl, 1 M)	12	60	5	95	1	45	47	92	5	40	80	81

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
