# Peer review of "Analysis and Optimization of Mn Removal from Contaminated Solid Matrixes by Electrokinetic Remediation"

_ijerph, 2020, doi:10.3390/ijerph17061820_

Round 1

Reviewer 1 Report

This work has done an interesting work. My comments are as follows1. The Introduction section is too weak, research background, research progress of the present literature need to be studied in detail, so as to make the novelty more clear. Thus this section should be strengthened.
2. Does the particle size has effect on remediation efficiency? Why 3 μm of kaolinite was used ?
3. The species of Mn(II) should be identified, by using XRD, may be Mn(OH)2 can be characterized.
4. The Conclusion section should be rewritten. Particularly, the discussion of citric acid (line 362-370) should not put in this section. Please discuss the role of organic acids in a separate section.
5. The enhanced electrokinetic remediation efficiency by organic acid such as citric acid, oxalic, tartaric acid, EDTA etc. was not a new finding, since many literature can be found as follows. However, none of them was cited.Jensen, P. E., Ahring, B. K., & Ottosen, L. M. (2007). Organic acid enhanced electrodialytic extraction of lead from contaminated soil fines in suspension. Journal of Chemical Technology & Biotechnology: International Research in Process, Environmental & Clean Technology, 82(10), 920-928.
Gu, Y. Y., & Yeung, A. T. (2012). Use of citric acid industrial wastewater to enhance electrochemical remediation of cadmium-contaminated natural clay. In GeoCongress 2012: State of the Art and Practice in Geotechnical Engineering (pp. 3995-4004).6. References contains many self-cites.

Author Response

Dear editors and reviewers,

The authors appreciate the time and effort of the reviewers reading and commenting our manuscript. We include detailed responses to the reviewers’ comments below.

RESPONSE TO REVIEWER 1

  1. The Introduction section is too weak, research background, research progress of the present literature need to be studied in detail, so as to make the novelty more clear. Thus this section should be strengthened.

The Introduction section was revised and rewritten including new cites and references to previous studies on electrokinetic remediation

  1. Does the particle size has effect on remediation efficiency? Why 3 μm of kaolinite was used?

Kaolinite have been used as a model soil specimen in electrokinetic remediation from the very beginning of the development of this technology. The particle size was selected based in the availability kaolin product of the producer company. The idea was to model a low permeability soil, that it is why we selected the small particle size available from the CAVISA company.

  1. The species of Mn(II) should be identified, by using XRD, may be Mn(OH)2 can be characterized.

In our previous works we identified the chemical species formed during the electromigration of Mn2+. The precipitation of Mn2+ in the alkaline environment in the cathode forms Mn(OH)2, this compound is a white-yellow precipitate. The Mn hydroxide easily transformed into MnO(OH) with a characteristic brown color. Thus, the precipitation of Mn is clearly observed by a brown band in the white kaolinite or as a precipitate in the cathode chamber.

  1. The Conclusion section should be rewritten. Particularly, the discussion of citric acid (line 362-370) should not put in this section. Please discuss the role of organic acids in a separate section.

This section has been revised according your comments.

  1. The enhanced electrokinetic remediation efficiency by organic acid such as citric acid, oxalic, tartaric acid, EDTA etc. was not a new finding, since many literature can be found as follows. However, none of them was cited.

The text of this section have been revised and new cites and references were included

Jensen, P. E., Ahring, B. K., & Ottosen, L. M. (2007). Organic acid enhanced electrodialytic extraction of lead from contaminated soil fines in suspension. Journal of Chemical Technology & Biotechnology: International Research in Process, Environmental & Clean Technology, 82(10), 920-928.

Gu, Y. Y., & Yeung, A. T. (2012). Use of citric acid industrial wastewater to enhance electrochemical remediation of cadmium-contaminated natural clay. In GeoCongress 2012: State of the Art and Practice in Geotechnical Engineering (pp. 3995-4004).

  1. References contains many self-cites.

Reference section has been completed with new references from key papers on electrokinetic remediation

Reviewer 2 Report

The article entitled “Analysis and optimization of Mn removal from contaminated solid matrixes by electrokinetic remediation” deals with an interesting subject such as the influence of the electric field strength and its mode of application to the soil in the remediation results. I believe it should be considered for publication. However, I understand that some minor aspects need to be carefully revised before publishing.

Introduction:

Line 48. Please, complete the sentence “(pH, …)”.

In this work, authors have applied EKR to different solid matrices: kaolin and sludge. It is recommended to discuss the differences between these matrices (not only in the introduction section but also in the results sections, for example comparing energy consumption referred to metal removal rate).

Materials and methods:

Line 93. It is said that after 24 hours, the sorption of Mn to the kaolinite particles has taken place. Have the authors studied the influence of time with the sorption process (aging)? Authors should justify this sentence.

Line 110. Could the authors justifify the values of electric potential and the electric current intensity selected to carry out the experiments?

Line 125. Can authors clarify if the methods are the same?

Results:

Line 129: “Electrokinetic”

Line 273: “an acid”

Line 299: “is not favored”

Line 312: It could be interesting to discuss about the influence of facilitating agent in real contaminated matrixes.

Line 344: It is recommended to express the energy as specific energy taking into account the metal removal percentage.

Conclusions:

From my point of view, the conclusion section should be summarized.

Author Response

The article entitled “Analysis and optimization of Mn removal from contaminated solid matrixes by electrokinetic remediation” deals with an interesting subject such as the influence of the electric field strength and its mode of application to the soil in the remediation results. I believe it should be considered for publication. However, I understand that some minor aspects need to be carefully revised before publishing.

Introduction:

Line 48. Please, complete the sentence “(pH, …)”.

The sentence was revised and completed

In this work, authors have applied EKR to different solid matrices: kaolin and sludge. It is recommended to discuss the differences between these matrices (not only in the introduction section but also in the results sections, for example comparing energy consumption referred to metal removal rate).

The comment of the reviewer was included in the new version of the manuscript, in introduction and in results sections

Materials and methods:

Line 93. It is said that after 24 hours, the sorption of Mn to the kaolinite particles has taken place. Have the authors studied the influence of time with the sorption process (aging)? Authors should justify this sentence.

This sentence was rewritten. As the reviewer said, 24 h is not a period of time enough to consider aging on the contaminant in the solid matrix, so we remove the comment about the possible adsorption of Mn on the kaolinite since there is not practical evidence that a significant absorption took place in the 24 h. the aging of the contaminants is very important in soil remediation, but aging takes much more time. In this study we prepared the mixture one day, and the day after we started the consolidation and the experiment. No aging can be considered in this conditions. This is a common practice with samples contaminated in the lab. This has to be considered when the test conditions are used for real soil samples with aged contaminants.

Line 110. Could the authors justifify the values of electric potential and the electric current intensity selected to carry out the experiments?

The potential gradient and current intensity was selected based on previous studies of our group and other groups in electrokinetic remediation with model kaolinite samples.

Pamukcu, S.; Kenneth Wittle, J. Electrokinetic removal of selected heavy metals from soil. Environ. Progr. 1992, 11, 241-250.

Ricart, M. T.; Cameselle, C.; Lucas, T.; Lema, J. M. Manganese removal from spiked kaolinitic soil and sludge by electromigration. Sep. Sci. Technol. 1999, 34, 3227-3241.

Line 125. Can authors clarify if the methods are the same?

The text was revised to clarify that the methods for the model kaolinite sample and the industrial sludge were the same.

Results:

Line 129: “Electrokinetic”

Corrected

Line 273: “an acid”

Corrected

Line 299: “is not favored”

Corrected

Line 312: It could be interesting to discuss about the influence of facilitating agent in real contaminated matrixes.

The discussion was revised according the reviewer comment

Line 344: It is recommended to express the energy as specific energy taking into account the metal removal percentage.

The electric consumption was reported as absolute value (for the experiment in the conditions at lab scale) and referred to the mass treated soil/sludge. The energy consumption per mass of soil is interesting because it helps in the scaling up of the process. In the present study the electric consumption per metal removal is not informative because in real samples the removal of the first test was zero, so the ratio will be infinite.

Conclusions:

From my point of view, the conclusion section should be summarized.

The conclusions section was revised and the conclusions summarized.

Round 2

Reviewer 1 Report

Thank you for improve the manuscript.